



# The Impact of Ice Structures and Ocean Warming in Milne Fiord

Jérémie Bonneau[1], Bernard E Laval[1], Derek Mueller[2], Yulia Antropova[2], and Andrew K Hamilton[3]

[1]Department of Civil Engineering, The University of British Columbia, Vancouver, Canada
[2]Department of Geography and Environmental Studies, Carleton University, Ottawa, Canada
[3]Department of Earth and Atmospheric Sciences, The University of Alberta, Edmonton, AB, Canada

**Correspondence:** Jérémie Bonneau (jbonneau@mail.ubc.ca)

**Abstract.** Arctic tidewater glaciers and ice shelves are undergoing rapid attrition, with warmer ocean temperature playing an important role. However, the relationship between ocean temperature and ice structure retreat is complex and may change as the ocean warms and as the ice structure geometry evolves. In order to explore ice-ocean interactions and the impact of retreating ice structures in a glacial fjord, we use a numerical ocean model of Milne Fiord, which features an ice shelf and a tidewater glacier with a floating glacier tongue (part of which is detached). We model past, present and potential future ice configurations. Our results reveal that the average submarine melting is negligible across the ice shelf (<2 cm a$^{-1}$), but can dominate thinning rates (>20 cm a$^{-1}$) at specific locations where the ice is thick (>50 m) along the seaward edge. Our simulations also indicate that the temperature of water reaching the grounding line does not vary significantly when the ice shelf and glacier tongue are removed. In addition, we carry out a series of simulations with increasing ocean temperature which reveal a quasi-linear relationship between ocean temperature and submarine melting at the grounding line. Using this relationship and ocean temperature predictions for different greenhouse gas emission scenarios (2020 to 2100), we estimate that Milne Glacier will continue to retreat for at least 50 years, solely in response to ocean forcing. This study highlights the ongoing and future retreat of ice structures in a region considered as the Last Ice Area.

## 1 Introduction

Ice structures, here defined as large submerged glacier ice features, are currently experiencing a global retreat in the Arctic; 85% of marine-terminating glaciers (hereafter 'tidewater glaciers') in the Northern Hemisphere have retreated between 2000 and 2020 (Kochtitzky and Copland, 2022). During this period, many glaciers have also lost their ice tongues, including Jakobshavn (Motyka et al., 2011), Zacharie Istrom (Mouginot et al., 2015) and Yelverton glaciers (White and Copland, 2019). A warmer ocean is often considered a major cause of this phenomenon (e.g. Holland et al., 2008; Straneo et al., 2012; Cai et al., 2017; Millan et al., 2022). In the Southern Hemisphere, the intrusion of warm Circumpolar Deep Water inside ice shelf cavities has also been identified as a major cause of ice shelf retreat (e.g. Azaneu et al., 2023; Si et al., 2024).

It is an axiom that increased water temperature leads to more melting, but the exact relationship between the temperature of water offshore of glacial fjords or ice shelf cavities and the melting of ice structures is complex. This relationship depends on multiple factors such as the presence of a sill (Zhao et al., 2019, 2021), the presence of icebergs (Davison et al., 2020; Hager et al., 2024), ice shelf thickness and geometry (De Rydt et al., 2014; Bradley et al., 2022; Steiger et al., 2022), and the




amount of subglacial discharge (Cowton et al., 2015; Slater et al., 2019). For instance, for four similar tidewater glaciers in West Greenland, Rignot et al. (2016) found that the dependency of glacier melting on water temperature can vary by more than 50% (their Table S1). This suggests that simple parameterizations for ice melting as a function of water temperature derived for a specific site should be used with caution for other glacial fjords or ice shelf cavities, unless validated. Moreover, in addition

to the variability between sites, changes at a specific location, such as glacier retreat or a large calving event can potentially change the derived relationship by altering the ocean circulation and hydrography. This is important because it could create a positive feedback loop where warmer water leads to enhanced calving, which leads to more warm water intrusion. Two recent numerical studies on Antarctic ice shelves indicate that major calving events do not substantially change the overall average submarine melting rate (<15% change), but can lead to significant melt increases at specific locations (>100%) (Bradley et al.,

2022; Poinelli et al., 2023).

In this study, we evaluate the impact of changing ice structures (ice shelf, glacier tongue) and increasing ocean temperature on ice-ocean interactions in Milne Fiord. Milne Fiord is an ideal site because it hosts an ice shelf at its mouth in addition to a tidewater glacier at its head (separate ice structures), and because observational efforts in the last 15 years have enabled the development and validation of a high-resolution ocean model, which can be used to predict the melting/freezing of the

ice structures (Bonneau et al., 2024b). Here, the model is first used to evaluate the consequences of removing ice structures (calving) on the hydrography and submarine melting. Then, it is employed to estimate the impact of the forecasted ocean warming in the region on the submarine melting of the glacier.

## 2    Background on the Region

Milne Fiord (82.5°N, -81°W) is situated on the northern coast of Ellesmere Island in the Canadian High Arctic (Figure 1A).

This unique site hosts an ice shelf, a detached glacier tongue and a tidewater glacier. It is located at the center of the coastline of Canada's newly designated Tuvaijuituq Marine Protected Area. Tuvaijuittuq, meaning "the place where the ice never melts" in Inuktut, is projected to be one of the last areas with multiyear sea ice in the Northern Hemisphere (Jahn et al., 2024), thereby serving as a refuge for ice-dependent species (Vincent and Mueller, 2020). However, in spite of the predicted resilience of sea ice in this area, the ice structures (ice shelves, glacier tongues, glaciers) in Tuvaijuittuq are currently experiencing sustained

retreat (e.g. Mueller et al., 2017; Kochtitzky and Copland, 2022; White and Copland, 2019).

### 2.1    Milne Ice Shelf (MIS)

Using explorer reports from expeditions in 1875-1876 and 1906, Vincent et al. (2001) estimated that the north coast of Ellesmere Island was fringed by a ~8400 km² ice shelf at the beginning of the 20th century. This 'Ellesmere Ice Shelf' (unofficial name) is believed to have formed around 5500 years ago (England et al., 2017). Over the past century, the Ellesmere

Ice Shelf has almost completely disintegrated; it had an extent of 535 km² in 2015 (Mueller et al., 2017). The last relatively stable fragment of the Ellesmere Ice Shelf was in Milne Fiord, until 2020, when Milne Ice Shelf (MIS) calved and lost 43% (80 km²) of its area, including its thickest ice (~90 m thick). Before the calving, the mean thickness of MIS ~44 m (Figure





1C; Hamilton, 2016). Ice cores and remote sensing have revealed that MIS is a composite ice shelf, formed by the floating extension of glaciers and the accumulation of sea ice, nourished by basal accretion and snow accumulation (Jeffries, 1992; Mortimer et al., 2012; Richer-McCallum, 2015; White, 2019). Since 2011, the only glacier feeding MIS is Glacier 2 (Figure 1B; Mortimer et al., 2012)). Until 2020, MIS acted as a dam for surface runoff from the watershed, forming a freshwater layer floating above the seawater known as an epishelf lake (Hamilton et al., 2017; Bonneau et al., 2021; Veillette et al., 2011). Milne Fiord epishelf lake drained through a basal channel under the MIS, where the ice draft was approximately 8 m thick (Bonneau et al., 2021). Although the epishelf lake has dramatically thinned since the 2020 calving event, it is unclear at the moment if it remains perennial or has transitioned to a seasonal feature.

## 2.2 Milne Glacier

Around 1959, the floating extension of Milne Glacier disconnected from MIS and retreated up-fjord, creating a gap between the ice shelf and MGT (Jeffries, 1986). This gap, filled by the epishelf lake and covered by perennial lake ice, has been expanding ever since (Mortimer et al., 2012). Due to the presence of the ice shelf and perennial ice, icebergs do not exit the fjord, resulting in a fractured glacier tongue, similar to Thwaites Western Ice Tongue (e.g. Benn et al., 2022). Between 2009 and 2011, a large rift formed <1 km from the grounding line (Mueller et al., 2017; Antropova et al., 2024), separating MGT from the glacier. In 2018, MGT covered ~66 km$^2$. A recent study combining remote sensing and radar observations from 2011 to 2023 estimated that the grounding line of Milne Glacier retreated by 53 m a$^{-1}$ at the center of the glacier and by 124 m a$^{-1}$ on the western side (Figure 1B; Antropova et al., 2024)). Milne Glacier grounding line has a maximum depth around 190 m and is currently located on a retrograde slope (Figure 1C), suggesting it is vulnerable to marine ice sheet instability (Schoof, 2007; Antropova et al., 2024). The estimated ice flux from the glacier varied from 0.03 to 0.14 Gt a$^{-1}$ between 2011 and 2019 (Van Wychen et al., 2016; Millan et al., 2017; Wychen et al., 2020; Antropova et al., 2024). The associated glacier velocities ranged from ~20 m a$^{-1}$ to ~160 m a$^{-1}$ between 2011 and 2020 with the acceleration of up to ~160 m yr-1 between 2016 and 2020 (Van Wychen et al., 2016; Millan et al., 2017; Wychen et al., 2020; Antropova et al., 2024).

## 2.3 Milne Fiord Oceanography

The water column in Milne Fiord is composed of three distinct layers (Hamilton et al., 2021)(Figure 2A,B). The uppermost layer, which was the ~10 m thick epishelf lake, is warm (>0°C) and fresh (<1 g kg$^{-1}$)(Hamilton et al., 2017; Bonneau et al., 2021). The epishelf lake is ice-covered year round and extremely stratified, therefore isolating the sea water below from atmospheric forcing Bonneau et al. (2021). Below the epishelf lake is a ~100 m thick layer of Polar Water near the freezing point. Below this layer is the warmer (>0°C) and saltier Atlantic Water. Because of the presence of MIS, freshwater from surface runoff, subglacial discharge and submarine melting accumulates in the upper 50 m of the water column (Hamilton et al., 2021; Bonneau et al., 2024b). A ~230 m sill restricts the renewal of deep water, but has limited impact on water reaching the glacier because it is deeper than the grounding line (Hamilton et al., 2021; Bonneau et al., 2024b). The results from a multiyear numerical simulation of the circulation in Milne Fiord, validated with observations, has revealed that the circulation is weak, but highly unsteady: three circulation modes with different patterns alternate in response to density variations on the



coastal shelf (Bonneau et al., 2024b). The water on the coastal shelf, which flows westward, originates mainly from the Canada Basin (Hamilton et al., 2021; Timmermans and Marshall, 2020). Except for intermittent leads along the coast in July, August and September (Figure 1A), the offshore region is covered by multiyear sea ice, land-fast most of the year (November-June).

## 3 Methods

### 3.1 Numerical Model

This study relies on a numerical ocean circulation model (MITgcm; Marshall et al. (1997)) of Milne Fiord developed and validated by Bonneau et al. (2024b). The model covers the extent of the fjord using a 150 m by 150 m horizontal grid (Figure 1A). The numerical domain extends 25 km east, west and north offshore from the fjord's mouth, with a telescoping grid. In the vertical, the cells are 2 m for the first 50 m and then linearly extend to 30 m at 600 m depth. Vertical eddy viscosity and diffusivity are set to $10^{-5}$ m s$^{-2}$ and $10^{-6}$ m s$^{-2}$, respectively. Horizontal eddy viscosity is determined using the Smagorinsky method (Smagorinsky, 1963) (coefficient=2.0) and the horizontal diffusivities are set to 3.0 m s$^{-2}$. To model the heat, salt and momentum flux at the boundaries with the ice structures, we take advantage of the separation of the glacier tongue from the glacier in 2009-2011. The ice shelf and the glacier tongue are modelled using the Shelfice package (Losch, 2008) with a drag coefficient ($C_d$) automatically determined so that $u$ and $v$ are zero at the interface (corresponding to a no slip boundary condition). At the glacier face, the fluxes are determined with the IcePlume package (Cowton et al., 2015)($C_d$=0.01, heat transfer coefficient $\Gamma_\Theta$=0.044, salt transfer coefficient $\Gamma_S$=0.00124, minimum background velocity $v_{hor}$=0.035 m s$^{-1}$), modified for a glacier with a constant overhanging slope of 4:1 (Bonneau et al., 2024b). For all simulations, the morphology of the ice structures is held fixed (melting/freezing does not change the ice thickness). The amount of subglacial discharge is determined by integrating the negative surface mass balance from RACMO2.3 (Noël et al., 2018). Offshore, the model is forced with temperature, salinity and velocities from ORAS5 reanalysis (Zuo et al., 2019). To reproduce the epishelf lake, which recharges every summer because of surface runoff, the upper 3 grid cells were relaxed towards mooring observations (temperature and salinity). See Bonneau et al. (2024b) for a more detailed description of the model and its validation and Bonneau et al. (2024a) for a discussion of the mechanism transferring heat from the open ocean to the glacier face.

### 3.2 Numerical Experiments

This study is divided into two main parts. The first part explores the impact of the ice structures in Milne Fiord on water properties and melting/freezing of these ice structures. For this, the numerical model is run with four different ice structure configurations (Figure 1A). The first ice configuration is the same as Bonneau et al. (2024b) and represents Milne Fiord before the July 2020 MIS calving event; it is denoted $pre2020$. The second part represents the fjord as of 2024; it is denoted $now2024$. The third ice configuration represents the fjord without an ice shelf; it is denoted $nois$. The fourth configuration represents the fjord without an ice shelf and without a glacier tongue; denoted $nogt$. $nois$ and $nogt$ are configurations expected to arise in



the coming decades, as has happened for other fjords along the northern coast of Ellesmere Island over the last two decades (Copland et al., 2007; Mueller et al., 2003; White and Copland, 2019; Mueller et al., 2017). For the *nois* and *nogt* runs, there is no temperature and salinity relaxation in the three uppermost grid cell to reproduce the epishelf lake was not. For each configuration, the model is spun up for 0.5 year starting in May 2011 and the last 3 years (November 2011 to November 2014) are used for analysis. A 3-year simulation is sufficient to obtain accurate (<20% error) average along-fjord heat and salt fluxes as well as accurate mean circulation patterns Bonneau et al. (2024b).

The second part of this study explores the impact of warming ocean temperatures on the melting of the glacier face. For this purpose, the model is run with the *nogt* ice configuration and with four different temperature increases (added to the baseline *nogt* ORAS5 offshore forcing). The temperature increases were determined using a multimodel mean from the fifth and sixth Coupled Model Intercomparison Project (CMIP). The four ocean warming simulations ($T03$, $T09$, $T16$ and $T30$) are intermediate steps in different greenhouse gas emission scenarios (RCP2.6/SSP126, RCP4.5/SSP245 and RCP8.5/SSP585; S1 and Figure S1). The CMIP multimodel mean ocean temperature increase corresponding to a warming of 0.3, 0.9, 1.6 and 3.0°C at 220 m depth were added to the ORAS5 baseline offshore forcing (*nogt*). 220 m corresponds to the grid cell just above the sill and just below the grounding line of Milne Glacier. Above 220 m, the temperature increase decreases with height for all scenarios (Figure 2B). The amount of imposed subglacial discharge at the grounding line of Milne Glacier is the same for all cases (Figure 2C). For each of the ocean warming simulations, the model is run 2.5 years and the last two years are used for analysis. Two years are sufficient because the melting of the glacier face does not show high variability (Section 4.2.3), therefore this duration is sufficient to obtain an accurate average.

### 3.3 Water Properties Metrics

To characterize the water properties, the water column inside the fjord is divided into four distinct layers and the water property anomalies, as well as the differences between the simulation with different ice structure configurations are analysed. Temperature anomalies are defined as the temperature inside the fjord minus the temperature offshore ($\Theta - \Theta_{off}$). Salinity anomalies likewise ($S - S_{off}$). Unless stated or plotted otherwise, the water properties at the center of the fjord are used for $\Theta/S$ when calculating these anomalies. We also calculate the volume exchange between the coastal shelf and the fjord ($Q_{ex}$), the pycnocline upwelling inside the fjord ($\Delta z_\rho$), the fraction of submarine meltwater ($f_{SM}$) and the total amount of freshwater ($f_{SM+SD}$). $Q_{ex}$ is calculated using along-fjord velocities ($U$):

$$Q_{ex} = \frac{1}{2} \int |U| dA \tag{1}$$

where $A$ is the cross-section at the mouth of the fjord ($X$=33 km). The upwelling, $\Delta z_\rho$, is calculated as the average depth difference between isopycnals offshore and at the center of the fjord ($X$=20 km) for a specific layer. Below the settling depth of the subglacial discharge plume ($\sim$40 m), the fraction of submarine meltwater ($f_{SM}$) is estimated using the temperature difference between the fjord ($\Theta$) and offshore along isopycnals, such that ice melting produces a cold anomaly represented by the mixing of ambient offshore water ($\Theta_{off,\rho}$) with water at -92.5°C ($\Theta_{SM}$) to account for the latent heat of fusion (Hamilton





et al., 2021):

$$f_{off,\rho} + f_{SM} = 1 \qquad \cup \qquad f_{off,\rho}\Theta_{off,\rho} + f_{SM}\Theta_{SM} = \Theta \qquad \rightarrow \qquad f_{SM} = \frac{\Theta - \Theta_{off,\rho}}{\Theta_{SM} - \Theta_{off,\rho}} \tag{2}$$

where $f_{off,\rho}$ is the fraction of offshore water. $\Theta_{off,\rho}$ is taken at the same density level as $\Theta$, but offshore. Above the minimal

settling depth of the subglacial discharge plume, the combined fraction of submarine meltwater and subglacial discharge water

($f_{SM+SD}$) is estimated using the fact that both subglacial discharge and submarine melting waters have a negligible salinity

($S_{SM+SD} \approx 0$ g kg$^{-1}$). Therefore, $f_{SM+SD}$ can be determined using the salinity inside the fjord ($S_{obs}$) :

$$f_{off,z} + f_{SM+SD} = 1 \qquad \cup \qquad f_{off,z}S_{off,z} + f_{SM+SD}S_{SM+SD} = S \qquad \rightarrow \qquad f_{SM+SD} = 1 - \frac{S}{S_{off,z}} \tag{3}$$

The offshore salinity ($S_{off,z}$) is taken at the same depth as $S$ because upwelling is masked by the accumulation of freshwater

above 40 m depth. Since subglacial discharge entrains saltier deep waters and transport them upwards, (3) underestimates

$f_{SD+SM}$. Similarly, (2) also underestimates $f_{SM}$ below 40 m since the neglected subglacial discharge water and associated

upwelled deep water result in warm anomalies, offsetting part of the cold anomaly used to determine $f_{SM}$. Nevertheless, these

estimates satisfy the present objective of comparing water properties for the different ice structure configurations. The amount

of $f_{SM}$ and $f_{SM+SD}$ is reported in water height ($\int f dz$). For example, an average fraction of 0.01 over a 10 m layer is 10 cm.

### 3.4 Glacier Face Melting Parameterization

To compare the results to other modelled glacial fjords, the submarine melting parameterization of Rignot et al. (2016) is

employed:

$$M_{GL} = (Ahq_{sd}^{\alpha} + B)TF^{\beta} \tag{4}$$

where $M_{GL}$ (in units of m d$^{-1}$) is the horizontal submarine melting rate at the grounding line (undercutting). $h$ is the thickness

of the glacier at the grounding line (200 m), and $q_{sd}$ is the rate of subglacial discharge volume (in units of m$^3$ d$^{-1}$) divided by

the submerged area of the glacier face. $TF$ is the temperature above the local freezing point ($\Theta_f$), commonly referred to as the

thermal forcing ($\Theta - \Theta_f$). In this study, the $TF$ value used in (4) is the average $TF$ value from the sea surface to the grounding

line depth, following Morlighem et al. (2019). Using remote sensing observations to calculate the retreat of five tidewater

glaciers in West Greenland, Rignot et al. (2016) arrived at mean values for the coefficients $A$, $\alpha$, $B$ and $\beta$ of 3·10$^{-4}$, 0.39, 0.15

and 1.18, respectively. These coefficients have not been validated in other glacial fjords and differed by up to 50% between

fjords in the Rignot et al. (2016) study. The effect of the fjord's geometry and ocean forcing mechanisms (e.g. tides, coastal

trapped waves, buoyancy-driven circulation) are encapsulated in these coefficients. The more vigorous the fjord's circulation

and the exchange with the coastal shelf, the higher these coefficients should be. In this study, melting rates presented in m and

m$^3$ per unit of time are based on an ice density of 920 kg m$^{-3}$.

### 3.5 Glacier Retreat

To verify our glacier melt estimates and to predict the future retreat of Milne Glacier due to submarine melting, we focus on

the grounding line position at the center of the fjord. A thinning of the glacier at the grounding line ($\delta h$, negative for thinning)



will result in a retreat ($\delta x$, negative for retreat)(Thomas and Bentley, 1978; Rignot, 1998; Wood et al., 2021):

$$\delta h = \delta x \left[ \left( 1 - \frac{\rho_w}{\rho_i} \right) \alpha_b - \alpha_s \right] \tag{5}$$

where $\rho_w$ and $\rho_i$ are the density of seawater (1030 g kg$^{-1}$) and glacier ice (920 g kg$^{-1}$). $\alpha_b$ and $\alpha_s$ are the slopes at the surface (fixed at -0.02) and at the base (bed) of the glacier at the grounding line (0.02 in 2020, varies along glacier). Both $\alpha_b$ and $\alpha_s$ are negative when sloping upward up-glacier (Figure 1C). The thinning at the grounding line ($\delta h$, negative for thinning) is function of the submarine melting at depth (undercutting, $M_{GL}$, positive), the velocity of the glacier ($u_G$), the slope of the glacier face, 4:1 in this case (section 3,1), and the surface mass balance ($\delta h_s$, negative for ablation):

$$\delta h = (u_G - M_{GL})/4 + \delta h_s \tag{6}$$

In this study, $M_{GL}$ is the average melting at 187 m depth (closest grid cell above the grounding line). For a zero surface mass balance, if the undercutting is higher than the glacier velocity, the glacier will thin and the grounding line will retreat, and vice versa.

## 4  Results: The Impact of Ice Structures

### 4.1  Water Properties

The impact of removing ice structures on the hydrography varies greatly with depth (Figure 3). To characterize these differences, the water column is divided into 4 layers: 1) the epishelf lake (0-10 m), 2) the subglacial discharge accumulation layer (10-40 m), 3) an intermediate layer affected by the ice shelf and glacier tongue (40-90 m), and 4) a deep water layer between the intermediate layer and the sill (90-220 m).

### 4.1.1  0-10 m: Epishelf Lake

The $\sim$10 m deep epishelf lake at the top of the water column is a markedly different layer with fresh (<1 g kg$^{-1}$ and warm (>-1°C) anomalies ($\Theta - \Theta_{off}$, $S - S_{off}$, Figure 3 and 4A,E,I,M). The 2020 MIS calving event results in a shorter basal channel allowing for a doubling of the outflow from the epishelf lake (Table 1). This results in a 0.12 m thinning of the epishelf lake accompanied by a 0.5 g kg$^{-1}$ salinity increase. This small difference is because the increased exchange only occurs below $S_A$=5 g kg$^{-1}$. When the ice shelf is completely removed, the epishelf lake disappears: salinity in this layer increases abruptly to seawater values (>29 g kg$^{-1}$) and temperature decreases below -1.3°C.

### 4.1.2  10-40 m: Subglacial Discharge Accumulation Layer

For the two runs with an ice shelf at the mouth of the fjord ($pre2020$ and $now2024$), the subglacial discharge plume in Milne Fiord generally settles between 10 m and 40 m. This results in warm and fresh anomalies within this layer (Figure 3 and 4A,I). The influence of subglacial discharge (0°C, 0 g kg$^{-1}$) is noticeable on T-S diagrams (Figure 3C,F) by water properties parallel to





the freezing line below 31 g kg$^{-1}$. Within this layer, the integrated amount of subglacial discharge and submarine melting is 2.9 m for the $pre2020$ run and 2.4 m for the $now2024$ run. This decrease in $f_{SD+SM}$ is attributed to enhanced volume exchange through the basal channel following the 2020 calving event (120 m$^3$ s$^{-1}$ after vs 90 m$^3$ s$^{-1}$ before). The decreased $f_{SD+SM}$ leads to saltier (0.5 g kg$^{-1}$) water in this layer. The complete removal of the ice shelf ($nois$, $nogt$) leads to unrestricted exchange with the coastal shelf resulting in water properties and stratification near offshore values (Figure 3). The unrestricted exchange leads to minimal amounts of subglacial discharge and submarine melting waters (<20 cm) and associated salinity increase (>1 g kg$^{-1}$). The temperature also decreases by more than 0.5°C following the calving event, indicating subglacial discharge and deep water entrained in the plume (both warm anomalies), not submarine melting (negative temeprature anomaly), dominate the temperature anomaly signal for the $pre2020$ and $now2024$ cases.

### 4.1.3 40-90 m: Intermediate Layer

Below the subglacial plume settling depth, from 40 m to 90 m (maximum MIS thickness before the MIS calving event), the $pre2020$ run shows a temperature anomaly of -0.08°C. The temperature in this layer increases as ice structures are removed (Figure 4A-D), indicating that trapped submarine meltwater is responsible for this cold anomaly. The amount of submarine meltwater at the center of the fjord is estimated at 4 cm for the $pre2020$ run. This decreases to 2.5 cm after the 2020 calving event and further decreases to ∼1 cm for the $nois$ and $nogt$ runs, further confirming trapped submarine meltwater is responsible for this anomaly. The removal of the ice shelf and the glacier tongue has the double effect of decreasing the amount of submarine meltwater (less ice available to undergo melting) and increasing the exchange with the coastal shelf, both acting to decrease the fraction of submarine meltwater, thereby generating the temperature difference between the runs. The positive ∼0.2 g kg$^{-1}$ salinity anomaly (Figure 4I), on the other hand, cannot be caused by trapped submarine meltwater, which would decrease salinity. Examination of along-fjord isopycnals reveals that the positive salinity anomaly is caused by upwelling. For the $pre2020$ run, the mean upwelling at the center of the fjord is 5.9 m. Together with the average vertical salinity gradient of 0.038 g kg$^{-1}$ m$^{-1}$, this upwelling yields a salinity anomaly of 0.22 g kg$^{-1}$, which is consistent with the salinity anomaly observed. The same analysis reveals a decrease in the amount of upwelling with 5.3 m, 4.3 m and 4.6 m for the $now2024$, $nois$ and $nogt$ runs, respectively, explaining the freshening observed when ice structures are removed. Upwelling should also result in a temperature increase, but this signal is masked by the accumulation of submarine meltwater (cold anomaly). Consistent with the change in temperature, the reduced upwelling is the consequence of enhanced exchange with the coastal shelf when the ice shelf is removed (Table 1).

### 4.1.4 90-220 m: Deep Layer

Between 90 m and the sill, cold temperature anomalies from submarine meltwater are present near the Milne Glacier grounding line for all four configurations (Figure 4A-D). This is also evidenced by a temperature depression, which moves the water closer toward the freezing temperature, moving parallel to the melting line on T-S diagrams (Figure 3E,F). Moving down-fjord, submarine meltwater is diluted with offshore water, increasing the temperature of this layer and allowing the upwelling signal (positive temperature anomaly) to prevail. For example, the submarine meltwater content in this layer decreases from 37 cm 2



**Table 1.** Volume exchange ($Q_{ex}$), isopycnal upwelling ($\Delta z_\rho$), subglacial discharge fraction ($f_{SD}$) and submarine meltwater fraction ($f_{SM}$) for the four layers and four ice structures configurations. The number following $\pm$ indicate one standard deviation.

|  | $pre2020$ | $now2024$ | $nois$ | $nogt$ |
|---|---|---|---|---|
| 0-10 m: $S$=5 g kg$^{-1}$ [m] | 7.4±0.1 | 7.3±0.1 | 0 | 0 |
| 0-10 m: $Q_{ex}$ [m$^3$ s$^{-1}$] | 23 | 45 | 350 | 290 |
| 10-40 m: $Q_{ex}$ [m$^3$ s$^{-1}$] | 90 | 120 | 2.0×10$^3$ m | 2.0×10$^3$ |
| 10-40 m: $\int f_{SD+SM}$ [m] | 2.9±1.0 | 2.4±0.8 | 0.1±0.1 | 0.1±0.1 |
| 40-90 m: $Q_{ex}$ [m$^3$ s$^{-1}$] | 0.88×10$^3$ | 1.3×10$^3$ | 1.5×10$^3$ | 1.7×10$^3$ |
| 40-90 m: $\Delta z_\rho$ [m] | 5.9±1.2 | 5.3±2.4 | 4.3±0.8 | 4.6±0.6 |
| 40-90 m: $\int f_{SM}$ [m] | 0.040 ±0.015 | 0.025±0.015 | 0.010±0.015 | 0.011±0.017 |
| 90-220 m: $Q_{ex}$ [m$^3$ s$^{-1}$] | 2.8e×10$^3$ | 2.9×10$^3$ | 2.2×10$^3$ | 2.3×10$^3$ |
| 90-220 m: $\Delta z_\rho$ [m] | 3.5±0.9 | 3.4±1.4 | 2.1±0.8 | 1.7±0.7 |
| 90-220 m: $\int f_{SM}$ [m] | 0.11±0.05 | 0.09±0.04 | 0.14±0.03 | 0.14±0.03 |

km away from the grounding line to 11 cm 18 km away from the grounding line. As the ice structures are removed, the amount of upwelling decreases from 3.5 m ($pre2020$) to 3.4 m ($now2024$) to 2.1 m ($nois$) to 1.7 m ($nogt$). This results in a cooling (Figure 4F-H) and freshening (Figure 4N-P) of the 90-220 m layer.

### 4.1.5 Comparison to Observations and Variability of the Water Properties

Observations from field campaigns carried out in July 2012 to 2019, 2022 and 2023 agree generally well with model values of $f_{SM}$, $f_{SD+SM}$ and $\Delta z_\rho$ (90-220 m layer) (Figure S2). On the other hand, observed values for $\Delta z_\rho$ (40-90 m layer) are generally lower than modelled values. This is in part because the observed subglacial discharge settling depth is $\sim$15 m lower

than simulated. The most significant cause of this discrepancy however, is likely the high temporal and spatial variability in the observations.

Overall, our modelling reveals that hydrographic changes in Milne Fiord following the 2020 calving event are minimal (Figure 3D-F), and well within the range of observed and modelled variability (Figure S2 and S3). Without the exact same boundary conditions, these differences would be impossible to detect. Accordingly, observations after the MIS calving event do not reveal

significant hydrographic changes compared to before (Figure S3), except at the very surface (the thinning of epishelf lake). This minimal difference due to the 2020 calving event is consistent with the remaining ice shelf which is still $\sim$40 m thick and extends across the width of the fjord.





## 4.2 Melting of the Ice Structures

### 4.2.1 Milne Ice Shelf (MIS)

The model shows MIS experiences both basal melting and freezing before and after the 2020 calving event (Figure 5A,B). Melting/freezing rates are highly variable in both space (Figure 5A,B) and time (Figure 6A); they vary from $15 \times 10^3$ m$^3$ d$^{-1}$ of freezing (equivalent to an area average thickening of 0.06 m a$^{-1}$) to $75 \times 10^3$ m$^3$ d$^{-1}$ of melting (equivalent to 0.46 m a$^{-1}$ for the $pre2020$ run and to 0.80 m a$^{-1}$ for the $now2024$ run). Generally, melting occurs over areas where the ice draft is greater than ~40 m while freezing occurs in shallower areas. This is consistent with water temperature which tends to be near the freezing 270 point (within 0.1°C) from 10 m to 40 m (Figure 3).

While the depth predominantly controls whether melting or freezing occurs, the magnitude of the currents under MIS ($|U|$) is responsible for periods of enhanced melt ($M$) (Figure 6B, $R^2$ of 0.89 between spatially averaged $M$ and $|U|$). Consequently, the highest time averaged melting rates (0.4 m a$^{-1}$) are found under the thicker seaward areas of the cie shelf where currents are stronger. Conversely, the maximum freezing rates are located on the west side of the ice shelf in thinner areas where water 275 is trapped and therefore prone to enhanced vertical heat loss through the ice shelf as lateral exchange is not possible. The high temporal variability of melt rates under MIS driven by currents is consistent with the unsteady circulation described by Bonneau et al. (2024b).

The 2020 calving event results in a marked increase of melting along the new ice shelf front and around the thicker ice on the west side at $X$=27 km where Glacier 2 feeds MIS. However, it does not impact areas of freezing (ice draft <40 m, Figure 5D). 280 This increase in melting is due to stronger currents now reaching the remaining ice shelf (Figure 6B) and the warming of the 40-90 m layer inside the fjord (Figure 4F, Figure 6C), suggesting a weak positive feedback where calving leads to more basal melting, thereby further weakening the ice shelf.

The temporal and spatial average melting rate over the ice shelf is 0.01 m a$^{-1}$ for the $pre2020$ run and 0.005 m a$^{-1}$ for the $now2024$ run. This is negligible compared to the temporal and spatial average surface mass balance of 0.30 m a$^{-1}$ from 2008 285 to 2017 (White, 2019). However, similarly to the basal mass balance, the surface mass balance of MIS is spatially variable and has a strong along-fjord gradient, with average ablations of 0.16 m a$^{-1}$ over the down-fjord half of the ice shelf ($X$>32 km) and 0.55 m a$^{-1}$ over the up-fjord section (White, 2019). Therefore, basal melting can be an important process where the ice shelf is thick (>50 m). On the down-fjord section of the ice shelf, it can be the dominating thinning mechanism.

The submarine melting/freezing rate estimates from the $pre2020$ run have a similar spatial pattern to estimates using a sim- 290 plified two-equation parameterization and hydrographic observations (Hamilton, 2016). However, melt rates in this realistic numerical simulation are, on average, 10 times less than those estimated by Hamilton (2016) and include negative values (not permitted by the Hamilton (2016) model). By combining the surface mass balance gradient from White (2019) with the basal melt rate estimates from the $pre2020$ run, we obtain an average ice shelf thinning rate of 31 cm a$^{-1}$, which closely matches the 29 cm a$^{-1}$ obtained by Mortimer et al. (2012) for the 1981-2009 period (Figure 5E). This correspondence supports the low 295 rates of submarine melting obtained in this study and suggests that Hamilton (2016) employed drag and transfer coefficients values that were likely too high and overestimated $|U|$. The currents they employed were 2-8 times higher than those modelled



here. Furthermore, the spatial distribution of thinning over MIS obtained by Mortimer et al. (2012) aligns well with our results (Figure 5E), providing additional confidence in the modelled basal melt rates.

### 4.2.2  Milne Glacier Tongue (MGT)

The basal melting of MGT shows a similar spatial pattern to MIS, with freezing above an ice draft of 40 m and melting below (Figure 5A). On the other hand, contrary to MIS, the melting of MGT has a clear seasonal pattern, driven by velocity and temperature increase during and following periods of subglacial discharge (Figure 6D-F). Nonetheless, the melting/freezing rates are very small (-0.1 m a$^{-1}$<$M$<0.1 m a$^{-1}$). The spatially averaged rate for the entire simulation is equivalent to 0.02 m a$^{-1}$ of freezing for the $pre2020$ run. This is negligible compared to the surface mass balance (melting rate of ∼0.55 m a$^{-1}$; Figure 5A,E; White, 2019). The negligible basal melting rates obtained over most of MGT are in agreement with remote sensing estimates from 2011 to 2015 (Hamilton, 2016). However, these melting rates at the bottom of the keel are perhaps too low, likely because of the very low currents there (Figure 6E) and because ShelfIce does not take into account the ice slopes, which are known to enhance submarine melting (Rosevear et al., 2024)

The partial removal of the ice shelf ($now2024$) has no significant effect on the basal melting rates of MGT, current magnitude and thermal forcing (Figure 6D-F). The complete removal of the ice shelf ($nois$) results in an increase in current magnitude by 20%, and a decrease in thermal forcing by 23% because of stronger exchange with the coastal shelf (Section 4.1). The net effect is a 70% increase of the freezing rates, which overall are still negligible (3 cm a$^{-1}$).

### 4.2.3  Glacier Face

The melting of the glacier face in the $pre2020$ run was already discussed by Bonneau et al. (2024a). The main takeaways are that, although melting increases by ∼15% in summer due to subglacial discharge, most of the melting (∼85%) occurs outside of the subglacial discharge plumes and does not exhibit a strong seasonality. The average melting rate of the glacier face is 33 m a$^{-1}$, with rates of ∼100 m a$^{-1}$ at the depth of the grounding line (Figure 7A).

With a surface slope of -0.02, a basal slope of 0.02 (Figure 1C) and a grounding line retreat of 53 m a$^{-1}$ (Antropova et al., 2024), (5) yields a thinning rate $\delta_h$ of 0.93 m a$^{-1}$. Combined with ablation stake measurements closest to the grounding line showing a negative surface mass balance ($\delta h_s$) of ∼-60 cm a$^{-1}$(White, 2019) and an undercutting rate ($M_{GL}$) of 87.2 m a$^{-1}$, (6) yields an average glacier velocity of 85.9 m a$^{-1}$, which compares well with the observed surface velocities (20 to 160 m a$^{-1}$; Van Wychen et al., 2016; Millan et al., 2017; Wychen et al., 2020; Antropova et al., 2024), validating the submarine melt rates obtained from this numerical model.

The results from the four model runs with different ice structure configurations show that the presence of MIS and MGT has a very limited impact on the melting of Milne Glacier face (Figure 7). Melting rates from the $now2024$ run are the same as the melting rates from the $pre2020$ run (within 0.5%). The complete removal of the ice shelf ($nois$ run) leads to a 9% decrease in melting. This decrease is due to the cooling and freshening of the 90-220 m layer (Figures 3 and 4), which decreases the



thermal forcing by ∼0.05°C. The further loss of the glacier tongue partly offsets this cooling, resulting in an increase in melting rate for the *nogt* run compared to the *nois* run, although in both cases the melting rate is lower than before the calving event (*pre*2020 run).

## 5  Results: Consequences of Ocean Warming

According to our simulation with different ice structure configurations, the presence of MIS and MGT has a negligible impact on the melting rate of the glacier face ($\leq 9\%$; Figure 8). Therefore, the results of this study of the impact of ocean warming on the melt rate of Milne Glacier are applicable regardless of the state of the ice shelf and glacier tongue.

The time average $TF$ ($\Theta - \Theta_f$) is 0.76°C, 0.84°C, 1.2°C, 1.5°C and 3.1°C for the *nogt*, $T03$, $T09$, $T16$ and $T30$ runs, respectively. The results show that a $TF$ increase of 0.08°C ($T03$) has limited impact (total melt over the glacier face increases

by 8%), but that further warming has a more substantial impact: 38% increase for $T09$, 141% increase for $T16$ and 420% increase for $T30$ (Figure 8). For the five simulations, melt rates increase by a similar proportion during summer: up to ∼25% for summer 2012 and up to 10% for summer 2013, i.e. linear response to subglacial discharge. We note a possible residual effect of subglacial discharge on melting as elevated values of submarine melting starting at the onset of subglacial discharge persist ∼3 months after subglacial discharge stops.

Calculating the submarine melting rate at the grounding line ($M_{GL}$) as the average melting at 187 m and comparing it to the parameterization of Rignot et al. (2016) (4) reveal substantial discrepancies between this parameterization and our results (>factor 2, Figure S4). Therefore, the coefficients of (4) were determined using the five ocean warming simulations. The expression obtained for $M_{GL}$ for Milne Fiord is:

$$M_{GL} = 0.31 TF^{0.90} \tag{7}$$

The main difference between our parameterization (7) and the one from Rignot et al. (2016) (4) is the exclusion of the subglacial discharge term ($q_{sd}$). This is justified by our optimization (least square), which yields a better correlation between modelled and parameterized $M_{GL}$ when $A$ is set to 0. This is explained by the negligible increase of undercutting ($M_{GL}$) during summer (Figure S4); unlike the total amount of melting over the whole glacier face (e.g. Figure 6G, 8F), $M_{GL}$ does not exhibit a seasonal cycle. We attribute this major difference with the parameterization proposed by Rignot et al. (2016) to the different

numerical setup: Rignot et al. (2016)(using the model from Xu et al. (2013)) modelled a 150 m wide glacier with two subglacial discharge outflows, while we modelled a 4000 m wide glacier with two outflows. Therefore, the proportion of the glacier face affected by subglacial discharge plumes at depth is ∼ 27 times lower in our case, resulting in a negligible dependence of $M_{GL}$ on subglacial discharge when averaged over the width of the glacier.



## 6    Discussion

### 6.1    Ice Cover

This is the first modelling study that we are aware of to examine the impact of ice shelf or glacier tongue calving on water properties and submarine melting rates in a glacial fjord. Our results show that the removal of ice structures in Milne Fiord has a limited impact on water properties in the fjord. The exception to this is the surface water (above 15 m depth), which become considerably saltier following the complete removal of the ice shelf and drainage of the epishelf lake. This will lead to sea ice formation inside the fjord (instead of the formation of lake ice) resulting in occasional ice-free periods in summer as is seen in neighbouring fjords (e.g. Veillette et al., 2008). For example, since the calving of Ward Hunt Ice Shelf (80 km east; Figure 1A) in 2001, and the subsequent drainage of its epishelf lake, open water was visible in Disraeli Fiord 17 out of the 20 following summers. This is in stark contrast to the frequency of open water in Milne Fiord which has occurred only once during the same period.

### 6.2    Ice Shelf and Glacier Tongue Melting

The low simulated submarine melting rates under MIS and MGT (space and time average <0.02 m a$^{-1}$) are consistent with studies which identify atmospheric warming as the main cause for the loss of more than 75% of the ice shelf extent along Ellesmere Island over the last 60 years (White and Copland, 2019; Copland et al., 2017, 2007; Mueller et al., 2003). However, the comparison of the spatially varying basal melt rates obtained in this study with surface mass balance estimates reveals that submarine melting can, in fact, be the dominating thinning mechanism where ice is thick (>50 m) near the mouth of the fjord, where the surface mass balance is less negative because of lower summer temperature, higher precipitation and summer fog (White, 2019). We note observations of surface mass balance even closer to equilibrium near the seaward edge of Ward Hunt Ice Shelf (0.07 m a$^{-1}$ of melting; Braun et al., 2004). According to ORAS5 reanalysis (Copernicus Climate Change Services, 2021), the thermal forcing of the ocean above 100 m does not appear to have changed substantially between 1958 and 2019 (Figure S5). Considering the seaward portion of Ayles Ice Shelf (15 km to the east of Milne Fiord, calved in 2005) was ∼ 44 m thick (Copland et al., 2007) and that the seaward portion of Ward Hunt is ∼ 50 m thick (Braun et al., 2004), it appears that the Ellesmere Island Ice Shelf was sufficiently thick for submarine melting to be an important component of the mass balance. This leads to the hypothesis that submarine melting substantially contributed to the disintegration of the Ellesmere Island Ice Shelf by weakening its seaward edge.

The limited sensitivity of submarine melting rates to past ($now2024$) and future ($nois$) calving events in Milne Fiord is similar to the results obtained in a recent numerical study of Pine Island Ice Shelf (Bradley et al., 2022), where it was estimated the the retreat of the ice front by up to 50 km increased the spatial and temporal average ice shelf melt rate by less than 10%. Similar results where also found for Larsen C Ice Shelf for which numerical simulations indicate that the calving of iceberg A-68 (5800 km$^2$) did not substantially impact the spatial and temporal average ice shelf melt rate (Poinelli et al., 2023). However, similar to our results from the $now2024$ run, Poinelli et al. (2023) showed a local increase of melting along the new ice shelf front (Figure 4B,C). Moreover, even though the spatial and temporal average melt rates did not vary significantly following the





calving of A-68, Poinelli et al. (2023) calculated a doubling of melt rates around an important ice shelf pinning point, and noted that localized change in submarine melt rates can have an overall destabilizing impact. As MIS is only grounded on its sides, the pinning points are lateral and the most important one is likely where Glacier 2 (west side, X=27 km, Figure 1B) feeds the ice shelf. Our results show this is the second location where submarine melt rates increase the most following the 2020 calving event (∼75% increase), suggesting that enhanced local submarine melting may lead to further structural weakening.

## 6.3 Glacier Face Melting

Using two-dimensional numerical simulations of Petermann Gletscher, Cai et al. (2017) argued that the melt rates at the grounding line should increase following the removal of the ice shelf as this would lead to a steeper under-ice slope. We cannot confirm that this would be the case for Milne Glacier as the slope near the grounding line was kept constant in our numerical experiments. However, as our results show, water reaching the grounding line has similar temperature (within 0.15°C) and salinity (within 0.1 g kg$^{-1}$) regardless of the ice structure configurations, it is likely that local melt rates will increase if the slope becomes steeper. Consistent with our results, numerical simulations of Pine Island Ice Shelf (Bradley et al., 2022) and Larsen C Ice Shelf (Poinelli et al., 2023) with constant ice morphology also suggest that major calving events have limited impact on submarine melting along the grounding line, with more pronounced impacts near the seaward edge or along the main ocean intrusion pathways.

## 6.4 Future Submarine Melting-Induced Retreat of Milne Glacier

Using the $M_{GL}$ parameterization (7), a constant surface mass balance ($\delta h_s$=-0.60) and a constant glacier velocity ($u_G$=85.9 m a$^{-1}$, Section 4.2.3) calibrated with the observed grounding line retreat (53 m a$^{-1}$), we calculate grounding line retreat for three different greenhouse gas emissions scenarios (SSP126, SSP245 and SSP585, S1, Figure S1). For each of these scenarios, the ocean temperature (therefore $TF$) is determined by using the (time- and depth-varying) multimodel mean temperature increase. These estimates suggest that the grounding line of Milne Glacier will experience an accelerated retreat until 2030 and a continual near-constant rate of retreat of ∼200 m a$^{-1}$ during the following 30 years, regardless of the greenhouse gas emission scenario (Figure 10). After 2065, the low carbon emission scenario (RCP2.6/SSP126) leads to a stabilization of the glacier, while the moderate and high greenhouse gas emission scenarios result in an uninterrupted retreat. The rate or retreat of 200 m a$^{-1}$ obtained from 2030 to 2035 is similar to what was observed for Umiamako Isbrae in west Greenland (180 m a$^{-1}$ from 1989 to 2015; Rignot et al., 2016) and Petermann Glacier in northern Greenland (130-230 m a$^{-1}$ from 1992-2021; Millan et al., 2022). Varying the basal slope ($\alpha_b$) and the surface slope ($\alpha_s$) by ± 50% and increasing the surface mass balance ($\delta h_s$) by a factor of four does not significantly alter these results (Figure S6). However, these estimates do not include glacier dynamics. For instance, the MIS, MGT and perennial ice likely provided some buttressing to the glacier presently and in the past (Scambos et al., 2004; Mouginot et al., 2015, 2019). The loss of these ice structures could therefore lead to a speed-up of the glacier, resulting in thinning and further grounding line retreat (Joughin et al., 2021; Millan et al., 2022). The MIS, MGT and perennial ice also protect the glacier from ocean swell and currents (Glasser and Scambos, 2008; Massom et al., 2018). These processes could also lead to faster rates of retreat than what is estimated here. However, as the glacier retreats and



undercutting ($M_{GL}$) occurs at shallower depth, the effect of ocean warming will decrease (Figure 2B), potentially enhancing the impact of surface ablation and subglacial discharge, which will increase as air temperatures warm as well.

Our results show the vulnerability of the marine-terminating glaciers along the north coast of Ellesmere Island and are in line with the ongoing retreat of ice structures in the region (Mueller et al., 2017; White and Copland, 2018; White, 2019; Kochtitzky et al., 2022). The retreat of Milne Glacier to shallower depths also has ecological consequences as nutrients and phytoplankton

abundance differ between deep, shallow and land-terminating glacier fjords (Bhatia et al., 2021; Roberts et al., 2024).

## 7  Conclusion

In this study, we employed a numerical model validated by observations to explore ice-ocean interactions in Milne Fiord. To investigate the rapid changes currently happening in this system, we examined the impact of the loss of ice structures and ocean warming on the hydrography and submarine melting rates. The main results are:

1. Although the spatial and temporal average submarine melt rate under Milne Ice Shelf (<0.02 m a$^{-1}$) is negligible compared to surface mass balance (0.3 m a$^{-1}$), submarine melting is the dominant thinning mechanism locally where the ice shelf is >50 m thick near the ocean.

2. The loss of ice structures impacts the hydrography of Milne Fiord differently with depth. Above 15 m, the removal of the ice shelf results in a strong salinity increase (>15 g kg$^{-1}$). Below 15 m, the impact of the removal of ice structures
is limited to a temperature change of 0.15°C and salinity change of 0.2 g kg$^{-1}$, resulting in small changes in spatial and temporal average submarine melting/freezing rates. However, two specific locations now experience enhanced melting following the 2024 MIS calving event: the new ice shelf front and the basal surface of Glacier 2. These localized regions where melt rates show enhanced sensitivity to change in ice structure configuration suggest a weak positive feedback loop in which ice shelf calving leads to more melting which could promote more ice shelf calving.

3. The melting of the Milne Glacier face responds quasi-linearly to ocean warming. According to the analyzed CMIP5 and CMIP6 predictions, water offshore of Milne Fiord will warm by at least ∼0.2°C by 2040 regardless of the carbon emission scenario. Predictions based solely on changing submarine melting indicate a retreat of the Milne Glacier grounding line by 8 km by 2065. These predictions also suggest a subsequent stabilization for low greenhouse gas emission scenarios and further grounding line retreating for higher emission scenarios.

Situated at the center of Tuvaijuittuq's coastline, Milne Fiord is rapidly changing with the ongoing retreat of the ice shelf and glacier. While this study explores the impact ice structures and ocean warming on ice-ocean interactions, many processes were left aside, such as sea ice, glacier dynamics and warming air temperature. Investigating the impact of these mechanisms would provide a more complete portrait and improve the accuracy of the predictions.



*Code and data availability.* The mooring, CTD and ADCP data used to validate the model are available on the Polar Data Catalogue (Mueller
et al., 2021b, a; Hamilton et al., 2024; Mueller et al., 2024). The MITgcm inputs to run the simulation and the outputs are available on the
Federated Research Data Repository (Bonneau et al., 2024c)

*Author contributions.* JB built and validated the numerical model, analyzed the output, and prepared the manuscript. BEL funded the field-
work, and contributed to drafting the manuscript. DM designed, funded and participated in the fieldwork, as well as helped draft the
manuscript. YA participated in the fieldwork, contributed to drafting the manuscript, and conducted the remote sensing analysis of the
glacier used to validate the model and estimate future retreat. AKH designed and participated in the fieldwork and assisted with drafting the
manuscript.

*Competing interests.* The authors declare no competing interests

*Acknowledgements.* This project was supported by the following organizations: Killam Foundation (JB: UBC Doctorate fellowship 333),
The University of British Columbia (JB: 4YF-6569, 4YF-6456), Natural Sciences and Engineering Research Council of Canada (BEL: NRS-
2018-517975; RGPIN-2018-04843, DM: NRS-2011-408463; NRS-2016-408463; DG-2011-402314; DG-2016-06244), The Polar Continen-
tal Shelf Program (DM: 604-12; 626-13; 651-14; 642-15; 636-16; 647-17; 627-18; 651-19), ArcticNet GO-Ice (DM, JB, YA), ArcticNet
Freshwater Resources of the Eastern Canadian Arctic (DM), ArcticNet Impacts of the Changing Global Environment at Nunavut's Northern
Frontier (DM), Canada Foundation for Innovation (DM: 31410), The Ontario Research Foundation (DM: 31410), Polar Knowledge Canada
(JB, YB: NSTP), The Digital Alliance of Canada (DM: 4332-22)
The authors also thank the Polar Continental Shelf Program and all the individuals who took part in field campaigns from 2011 to 2019. We
also acknowledge T. Cowton and K. Zhao for providing the IcePlume package and B. Noël for providing the 1 km RACMO2.3 data.



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



**Figure 1.** Study Area. A: Earthview image of the northern coast of Ellesmere Island taken on August 18 2024. Ice shelf extent from Mueller et al. (2017). The black box is the extent of panel B and the blue box is the extent of the numerical model domain. The top right inset shows the location of Panel A. B: PlanetScope near-infrared image of Milne Fiord from August 18 2024 showing the extent of Milne Ice Shelf (MIS) and Milne Glacier Tongue (MGT). C: Along-fjord profile of ice thickness from ArcticDEM (Porter et al., 2018) and IceBridge (Paden et al., 2010) derived using hydrostatic equilibrium (for ArcticDEM data). Seabed elevation under the glacier is from IceBridge and seabed elevation under the ocean is from manual soundings (Bonneau et al., 2024b).

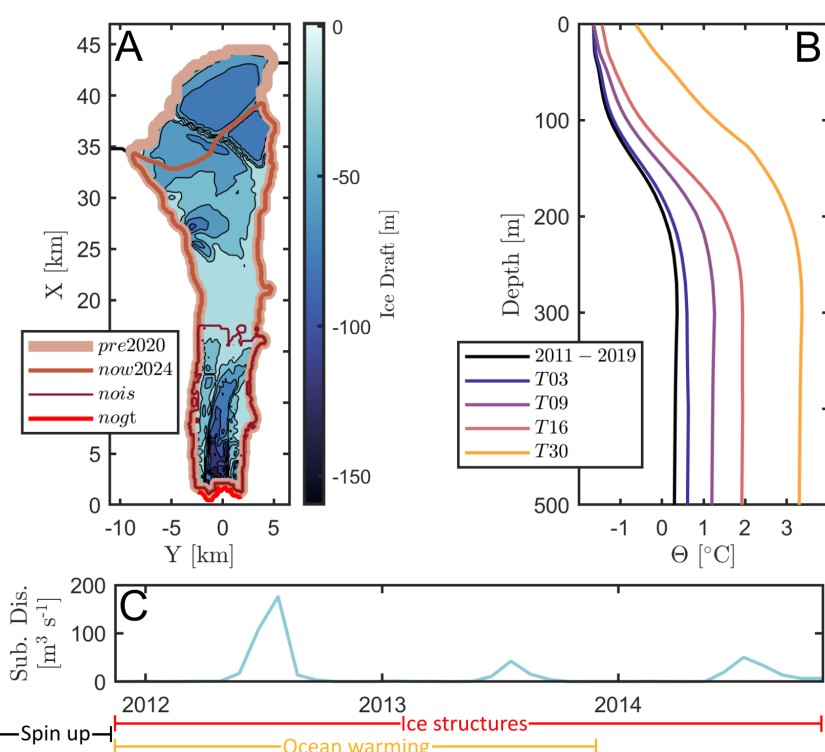

**Figure 2.** Model configuration. A: Ice draft and ice extent for the different ice structure scenarios. B: Mean offshore ocean temperature for the different ocean warming scenarios. C: Timeseries of amount of subglacial discharge (same for all scenarios).







**Figure 3.** Water properties from observations (left column) and model simulations (right column). A: July 2013 temperature profiles from offshore (dark blue line), at the center of Milne Fiord (cyan line) and near the grounding line (dotted cyan line). B: Modelled 3-year average temperature profiles at the center of Milne Fiord (solid lines) and near the grounding line (dotted lines) as well as average offshore temperature profile (black line). C: Same as A but for salinity. D: Same as B but for salinity. E: Temperature-Salinity plot from the three profiles in A and C. The dot-dashed line is the melting line; water involved in melting ice cools down parallel to this line. The negative slope line is the freezing point at the surface. The horizontal grey lines denote the four layers identified in Section 4.1. F: Same as E but for profiles in B and D. G: Density stratification for the three profiles in A and C. H: Same as G but for profiles in B and D.



**Figure 4.** Along-fjord temperature and salinity anomalies (relative to offshore) and differences (relative to the *pre*2020 run). Left column (A-D): Time average temperature anomalies ($\Theta$-$\Theta_{off}$) for the four ice configurations. E: Time average temperature for the *pre*2020 run. F-H: Time average temperature difference between the *now*2024, *nois*, *nogt* run and the *pre*2020 run. Third column (I-L): Time average salinity anomalies ($S$-$S_{off}$) for the four ice configurations. M: Time average salinity for the *pre*2020 run. N-P: Time average salinity difference between the *now*2024, *nois*, *nogt* run and the *pre*2020 run.



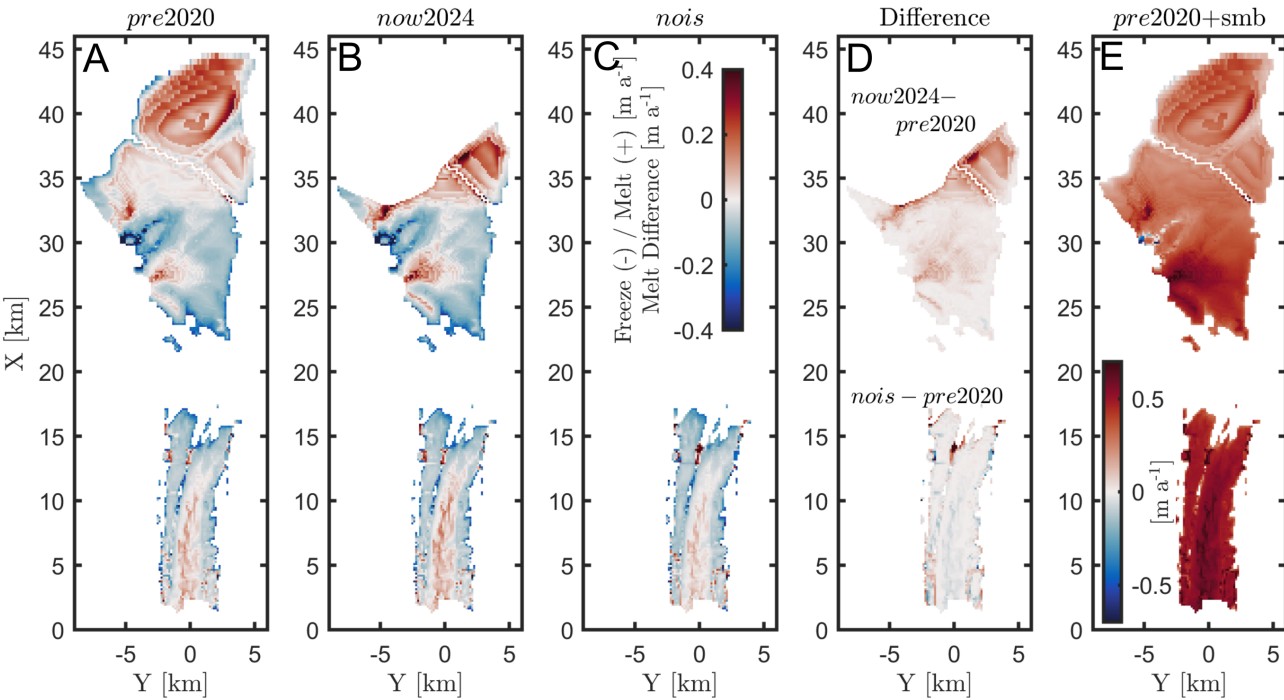

**Figure 5.** Time average basal melting and freezing of Milne Ice Shelf (MIS) and Milne Glacier tongue (MGT) from November 2011 to November 2014. A: Average melting (positive) and freezing (negative) rates for the $pre2020$ run. Colour scale in panel C. B: Same as A, but for the $now2024$ run. C: Same as A, but for the $nois$ run. D: Melt rate difference between the $now2024$ and the $pre2020$ run for MIS and between $nois$ and $pre2020$ for the MGT. Colour scale in panel C. E: Thinning rate for MIS and MGT using the surface mass balance from White (2019) and submarine melting from the $pre2020$ run (panel A). Note the different colourscale for Panel E (up to 0.6 m a$^{-1}$).





**Figure 6.** Timeseries of total melt rates ($M$), spatially averaged current speed ($|U|$) and spatially averaged thermal forcing ($TF = \Theta - \Theta_f$). For MIS and MGT, $|U|$ and $TF$ are from the cells directly under the ice. For the glacier face, $|U|$ and $TF$ are from an across-fjord cross-section 1 km down-fjord from the glacier. A-C are for Milne Ice Shelf (MIS), D-F are for Milne Glacier Tongue (MGT) and G-I are for the Milne Glacier face (MG). Grey shade denotes the periods with subglacial discharge.





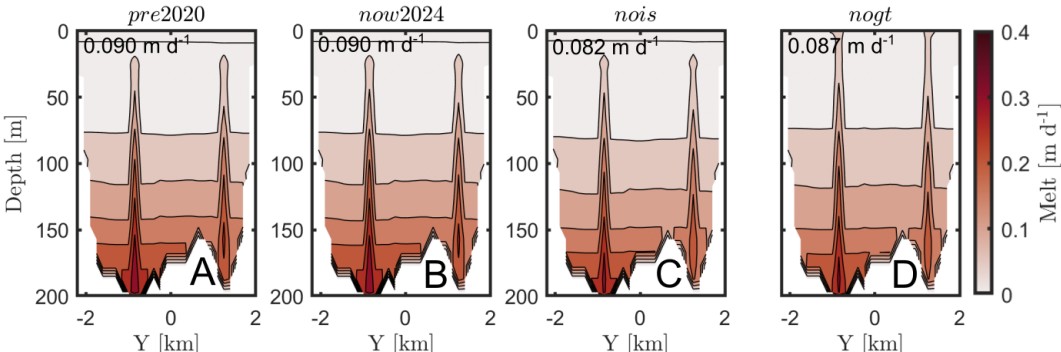

**Figure 7.** Timeseries of time average melting rate over the glacier face for the $pre2020$ (A), $now2024$ (B), $nois$ (C) and $nogt$ (D) simulations.

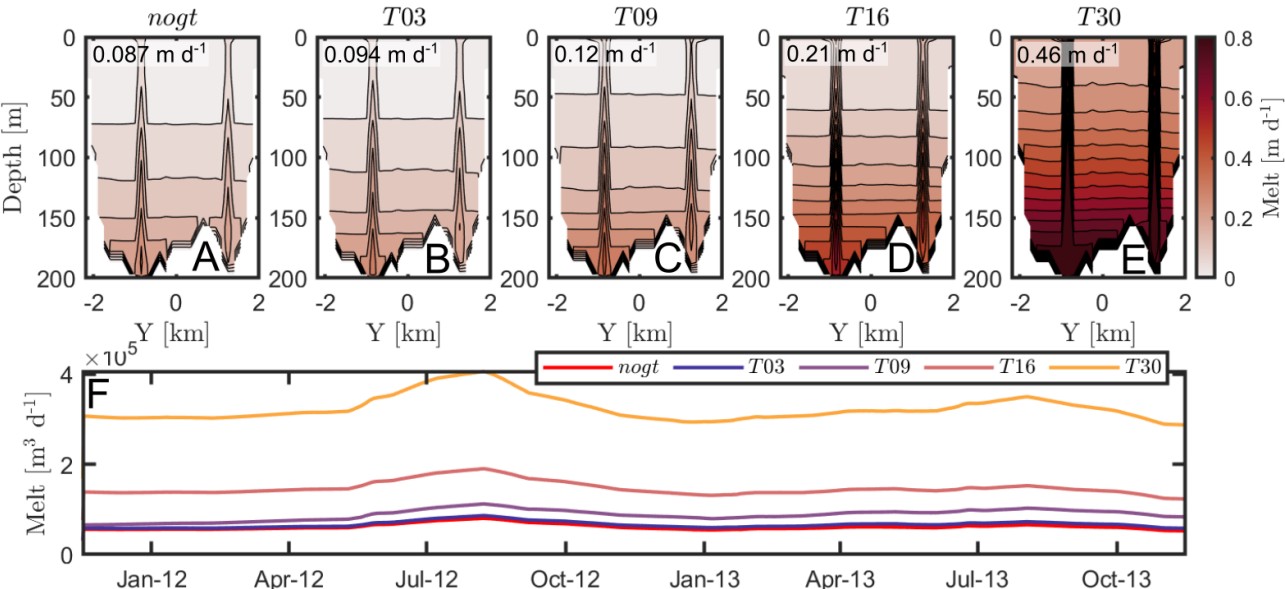

**Figure 8.** Melting rate of the glacier face for the five ocean warming scenarios (A-E, see subpanel title). The spatial and temporal average from November 2011 to November 2013 is in the top left corner of each panel. F: Timeseries of spatially integrated melting over the Milne Glacier face for the different temperature scenarios (see legend).



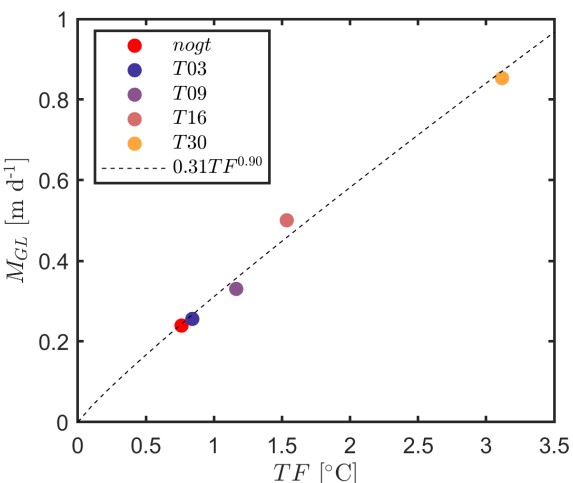

**Figure 9.** Undercutting from submarine melting ($M_{GL}$) as function of thermal forcing ($TF$) (7).



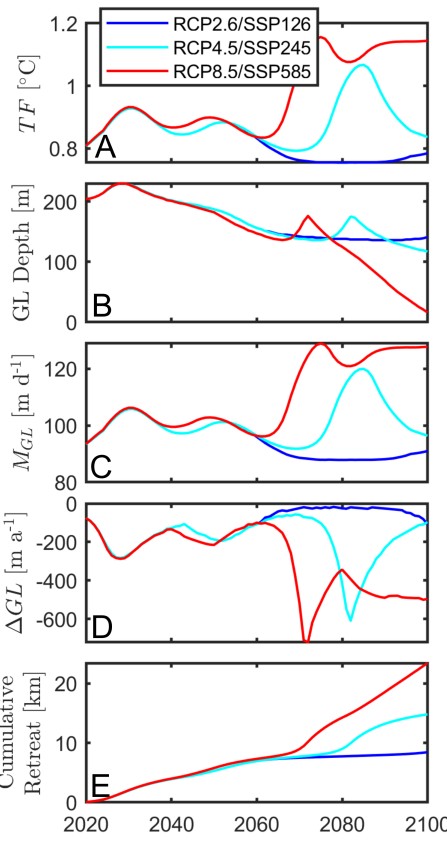

**Figure 10.** Submarine melting-induced retreat of Milne Glacier according to different carbon emission scenarios (see legend). Timeseries of A: Thermal forcing; B: Depth of the grounding line; C: Undercutting rate; D: Submarine melting-induced retreat rate of the grounding line; E: Cumulative retreat of Milne Glacier grounding line.