# Peer review of "The Impact of Ice Structures and Ocean Warming in Milne Fiord"

_EGUsphere, 2024_

## Author Comment (AC1)

Response to reviewer 1 for *The Impact of Ice Structures and Ocean Warming in Milne Fiord* by Bonneau et al. https://doi.org/10.5194/egusphere-2024-3651

We thank the reviewer for their kind words and their comments which improved the clarity of the manuscript. Most notably, we have added more comparison to Arctic studies, added a table with the numerical model's parameters and added more details on the subglacial discharge implementation.

Reviewer's comments in **dark bold**

Author's answers in blue

From the manuscript in *blue italic*

**Overview:**
**In this study, Bonneau et al. develop a high resolution (150 m) Milne Fiord configuration of the MITgcm to investigate the impact of changing ice-structure and hydrographic conditions on water column properties and basal/ice face melting rates. The model development builds on previously published work and the numerical experiments sample a wide breadth of ice structure configurations and boundary thermal forcing scenarios. In addition, the authors go the extra step to infer how warming ocean conditions based on CMIP5/6 output could impact retreat of Milne Glacier through the end of the century, tying the study together nicely. This is a really well crafted study and written paper. The numerical modeling is sound and the way the authors broke up the results section into the various depth levels was extremely helpful as a reader. I am also impressed with the quality of the writing as well as the rigor of the study (most times I thought "oh I wonder about XXX", you would cover it in the coming sections). Lastly, all of the conclusions are soundly backed up by the results and figures.**

**I do not have any major comments for the authors to address, but mainly include line comments and other small considerations that could improve the readability and quality of the manuscript. Once these are addressed, I would be delighted to see this wonderful study published in The-Cryosphere.**

**Line Comments:**
**● L18: There might be regional differences in the name of this glacier, but I believe Zacharie Istrom should be Zachariæ Isstrøm (please feel free to ignore this if I am wrong).**
Thank you for correcting us, this change was made.

**● L20: modified Circumpolar Deep Water (mCDW) enters Antarctic sub-ice shelf ocean cavities, whereas Circumpolar Deep Water is what exists off the continental shelf prior to it mixing with on-shelf water masses as it makes its way towards glacier tongues.**
Thank you for correcting us, this change was made.

**● L22: Increased water temperature \*in contact with the ice\* will lead to more melting**
Thank you for this clarification, this change was made.

**● L31-34: You mention Antarctic ice shelf - ocean studies, but are there any Arctic studies you could also mention here, either from Greenland or the Canadian Arctic?**
This is a good point, we have included Ryder, Petermann and 79N (Northern Greenland) to this paragraph.
Added: *A retrograde bathymetry, such as found in Petermann Fjord, is an example of feature potentially leading to enhanced submarine melting following glacier retreat, because of the increasing water temperature with depth (Millan et al. 2020). On the other hand, the emergence of a shallow sill during glacier retreat could obstruct the flow of warmer deep water to the grounding line, as is currently observed with Ryder and 79N Glaciers (Schaeffer et al. 2020, Jakobsson et al. 2020).*

**● L36-42: It might be useful early on in this paragraph to state where Milne Fiord is located (I know this is stated in the next section, but I think it would also be appropriate to provide it here as well).**
Of course, good point. We have added, in this paragraph, that Milne Fiord located on the northern coast of Ellesmere in the Canadian Arctic

**● L57: Change "the mean thickness of MIS ~44 m" to "the mean thickness of MIS was ~44 m"**
Good point, the change was made.

**● L74-76: In figure 1C, it looks like the grounding line is retrograde for maybe 1-2 km upstream of the 2011 grounding line; however, it seems like the 2013 grounding line may have already retreated down this retrograde slope and is now experiencing prograde bed topography, which would not be conducive for MISI. It might be worth specifying that the 2011 ice configuration was susceptible to MISI, but it is likely that its present-day configuration is not.**
We assume you mean 2023 and not 2013
This is a good point. The seabed data from IceBridge in Figure 1 is only a line near the center of the fjord. The bathymetry changes quite a bit across the fjord, so we cannot really say if it has now retreated to a prograde slope or not. Since we are not considering glacier dynamics in this manuscript, we removed the sentence about MISI altogether.

**● L95: It might be helpful to include a table that includes unique MITgcm coefficients used**
That is a good idea. We have included a new table in section 3.1 for this.

**● L124: Change "grid cell" to "grid cells". Also, I think this sentence ends wrong, check if this is a type.**
Yes, thank you for noticing.

**● L132: For the CMIP6 simulations, the proper emission scenario identifier is SSP1-2.6, SSP2-4.5, and SSP5-8.5. Can you include a citation for CMIP5/6. Lastly, it would be helpful to list the CMIP5/6 models that you used to compute your mean ocean forcing.**
We have added citation to the lead papers describing the CMIP5 and CMIP6 experiments. However, to not overload the reader, we prefer naming the six different models used in the supplementary information.

**● Eqn5-6, L187/193: This might be an obvious question, but do you combine these equations and solve for the retreat distance (delX) such that delX = (((u_g - M_{GL})/4)+delh_{s}) \**

**((1-rho_w/rho_i)\*alpha_b-alpha_s). Sorry for how horrible that equation looks. If this is the case, it might be worth ending this section by including that equation.**
Yes that is correct. We have added an equation combining these two expressions as suggested.

**● L213: When you reference the TS diagrams in figure 3, should the panels be E,F (and not C,F)?**
Absolutely, thank you for noticing.

**● L265-270: Do you know what causes the striations or discontinuities in melting in figure-5 (especially in the MIS in panels A/E)? My initial thoughts are that this is related to step-changes in the ice shelf draft that manifest in the melt, but I'm wondering if you have any insights into this?**
We assume the reviewer refers to the pattern highlighted here by the arrow. These discontinuities are definitely linked to step changes in the model (grid cell depth). But then there is more to it because these grid cell steps influence the circulation differently depending on their orientation with the mean ocean currents. A lee and "windward" thickness increase will have a different impact on the velocity field and therefore melting pattern.
We did not add anything about this in the manuscript.

[Figure]

**● L273: change "cie" to "ice"**
Yes, thank you for noticing this typo

**● L326-332: In Figure-7, it appears that the discharge-enhanced melt rates are able to reach the surface of the ice in the nogt experiment, but this is not the case for the other three experiments. This is interesting and might be worth noting in this paragraph. Is this because loss of the glacier tongue partially offsets cooling (L330), or is that unrelated to the vertical reach of the subglacial water? Also, perhaps I missed it in the methods section, but do you mention where along the glacier face you parameterize the subglacial discharge outflow locations?**
This is a good point. The subglacial discharge plumes are indeed reaching the surface in the nogt experiment. This is because of the lower density stratification in front of the glacier (Figure 3H). We have added: *A noteworthy change arising from the removal of the glacier tongue is the surfacing of the two subglacial discharge plumes (Figure 7d). This is due to the lower density stratification in front of the glacier (Figure 3h) because of the increased exchange with the rest of the fjord (fresher water not trapped behind the glacier tongue; Figure 3).*

We have added specifications related to the subglacial discharge outlets in the method sections:
*Two subglacial discharge outlets are used (in the IcePlume package), both coincident with depressions in the bathymetry (Section 4.2). While the exact number of subglacial discharge outlet is unknown, observations showed one outlet in a bathymetry depression in the west side of the fjord (Hamilton et al. 2021). Another one was therefore added in a depression on the other side of the fjord. Subglacial discharge is discharged equally in the two outlets.*

● **L386: Change "the the" to "that the"**

Yes, thank you for noticing this typo

● **L426: If the glacier is retreating onto more shallow bed topography, wouldn't that decrease the impact of subglacial discharge because it would be enhancing ice face melt over a shorter vertical distance?**

This is a good question. In all honesty, we don't know how melting related to subglacial discharge would respond to a shallower glacier. The goal of this sentence was to highlight that (negative) surface mass balance and subglacial discharge will also increase as the air temperature increases at the same time as water temperature. We have changed the sentence to:

*Finally, not accounted for in this study, air temperature increase will also result in a more negative surface mass balance and additional subglacial discharge, both accelerating the retreat.*

● **L430: It might be worth stating that validation of these parameterized grounding line retreat rates and how they are impacted by the ice dynamical feedbacks not considered can be validated in a numerical ice sheet modeling framework.**

That is a good point. We added: *Numerical modelling of Milne Glacier would allow the investigation of dynamical ice feedback and enable the validation of the grounding line retreat parameterization.*

**Figure Comments:**

● **Figure-1: For panel A, can you include descriptions of the black and blue lines in the legend?**

We are not sure which black/blue line the reviewer is referring to. The black box is the extent of panel B and the blue box is the extent of the numerical model domain, this information is included in the caption.

● **Figure-3: For panel F, it would be helpful to include a box in the main figure where the inset is taken from.**

This is a great idea. We added a box where the insets were taken from.

---

## Author Comment (AC2)

Response to reviewer 2 for *The Impact of Ice Structures and Ocean Warming in Milne Fiord* by Bonneau et al. https://doi.org/10.5194/egusphere-2024-3651

We thank the reviewer for their comments which improved the clarity and the reach of the manuscript. We have added more details on the implementation of the subglacial discharge in the model and have made a concerted effort to connect this study to other sites in the Arctic.

Reviewer's comments in **dark bold**

Author's answers in blue

From the manuscript in *blue italic*

**Reviewer 2**

**This paper investigates recent and anticipated future changes of the distribution of marine glacier ice in Milne Fiord. Using a combination of observations and ocean-circulation modelling, the authors examine how glacier retreat affects subsurface ice melt. The removal of the ice shelf and the ice tongue is found to have only a small impact on the melt at the grounding line of Milne Glacier.**

**The paper is well structured and written, and should be published after some minor revisions.**

**Comments**

**1) It would be useful to connect the present study to work in North Greenland (see e.g., Hill et al., 2017), as similar conditions prevail in this part of the Arctic. The paper cite many studies from Antarctica and Southern Greenland, but Petermann is the only glacier in North Greenland mentioned. In the discussion around L24, it would be relevant to cite Jakobsson et al. (2020) and Nilsson et al. (2023), who investigate the role of sills for blocking inflows of Antarctic Atlantic Water in north Greenlandic fjords. On L70 Thwaites is mentioned, but also C. H. Ostenfeld Glacier in North Greenland has a recently disintegrated ice tongue (Hill et al., 2017). And Ryder and C. H. Ostenfeld glaciers are situated in fjords that terminate in the perennially sea-covered Lincoln Sea (Hill et al., 2017).**

Thank you for this suggestion, it is a good idea to add more information and comparison to northern Greenland fjords. We have used the suggested studies and others.

We have added comparison to Ryder, 79N, Petermann Glaciers and other north Greenland tidewater glaciers in the introduction and discussion.

We have added comparison to Tracy, Steensby and C.H. Ostenfeld glacier tongues in the site description and discussion.

**2) Please describe the implementation of the subglacial discharge in the model in section 3.1; this issue is commented on around L355.**

We agree more details are warranted. We added on the location of the subglacial discharge outlets. This section now reads:

*The amount of subglacial discharge is determined by integrating the negative surface mass balance from RACMO2.3 (Noel et al. 2018). Two subglacial discharge outlets are used (in the IcePlume package), both coincident with depressions in the bathymetry (Section 4.2). While the exact number of subglacial discharge outlet is unknown, observations showed one outlet in a bathymetry depression on the west side of the fjord. Another one was therefore added in a depression on the other side of the fjord. Subglacial discharge is discharged equally in the two outlets.*

**3) Check the consistency of the dimensions in Eqs. (5, 6); or state that delta x and delta h are velocities (?)**

Good point, we have added the units (we here use m a$^{-1}$).

**4) Table 1, 90-220 m Q_{ex}: 2.8ex10^8 -> 2.8x10^8**

Yes, thank you

**5) L273: cie shelf  ->  ice shelf.**

Yes, thank you

References shared by the reviewer

Hill et al., 2017: A Review of Recent Changes in Major Marine-Terminating Outlet Glaciers in Northern Greenland. doi: 10.3389/feart.2016.00111

Jakobsson et al., 2020: Ryder Glacier in north-west Greenland is shielded from warm Atlantic water by a bathymetric sill. doi: 10.1038/s43247-020-00043-0

Nilsson et al., 2023: Hydraulic suppression of basal glacier melt in sill fjords. doi.org/10.5194/tc-17-2455-2023

**Citation**: https://doi.org/10.5194/egusphere-2024-3651-RC2